# Head and Neck Cutaneous Soft-Tissue Sarcoma Demonstrate Sex and Racial/Ethnic Disparities in Incidence and Socioeconomic Disparities in Survival

**DOI:** 10.3390/jcm11185475

**Published:** 2022-09-17

**Authors:** Muhammad Umar Jawad, Lauren N. Zeitlinger, Arnaud F. Bewley, Edmond F. O’Donnell, Sophia A. Traven, Janai R. Carr-Ascher, Arta M. Monjazeb, Robert J. Canter, Steven W. Thorpe, R. Lor Randall

**Affiliations:** 1Department of Orthopaedic Surgery, Good Samaritan Regional Medical Center, Corvallis, OR 97333, USA; 2Department of Orthopaedic Surgery, University of California-Davis, Sacramento, CA 95817, USA; 3Department of Otolaryngology, University of California-Davis, Sacramento, CA 95817, USA; 4Department of Medicine, University of California-Davis, Sacramento, CA 95817, USA; 5Department of Radiation Oncology, University of California-Davis, Sacramento, CA 95817, USA; 6Department of Surgery-General, University of California-Davis, Sacramento, CA 95817, USA

**Keywords:** cutaneous soft tissue sarcoma, disparities

## Abstract

Background: Cutaneous soft-tissue sarcoma (CSTS) of the head and neck are rare and are known to have aggressive clinical course. The current study utilizes a population-based registry in the U.S. to characterize these malignancies and explore disparities. Methods: National Cancer Institute’s (NCI) Surveillance, Epidemiology and End Result (SEER) database from 2000 to 2018 was queried to report incidence and survival data in 4253 cases in the U.S. Results: Males were 5.37 times more likely and Non-Hispanic-White people (NHW) were 4.62 times more likely than females and Non-Hispanic-Black people (NHB) to develop CSTS of the head and neck. The overall incidence was 0.27 per 100,000 persons in 2018, with a significant increase since 2000. Advanced age and stage, histologic group other than ‘fibromatous sarcoma’ and lower SES groups were independent factors for worse overall survival. Conclusions: CSTS of the head and neck demonstrate sex and racial/ethnic disparities in incidence and socioeconomic disparities in overall survival. Level of evidence: II.

## 1. Introduction

Cutaneous soft tissue sarcomas (CSTS) of the head and neck are rare [1] and represent a wide array of histopathological diagnoses [2]. CSTS of the head and neck often have an aggressive clinical course relative to other cutaneous malignancies [2]. They may be associated with significant cutaneous and systemic manifestations, and may have dramatic impacts on quality of life (QoL) [2]. Previous studies in the literature have reported the experience of single centers with resultant limited sample sizes [3,4,5] and, given the rarity of the disease, have been limited in their statistical power to analyze prognostic factors. In contrast, existing population-based studies have focused on cutaneous sarcoma located anywhere in the body [6] or aggregated soft tissue and bone sarcomas of the head and neck for combined analysis [7,8]. An Australian study addressing head and neck sarcoma has provided a sub-analysis of CSTS [9]. Thus, a dedicated population-based analysis of cutaneous soft tissue sarcoma (CSTS) of the head and neck in the US is lacking in the literature.

The National Institute of Minority Health and Health Disparities defines health disparities research as that which addresses health differences in socially disadvantaged populations related to specific outcomes [10]. In the context of cancer, disparities in the incidence, prevalence, rate of screening, stage at initial presentation, morbidity, survival, and financial burden of disease have been reported for multiple primary malignancies [11]. Disparity research exploring CSTS of the head and neck is lacking in the literature. We aimed to explore and report the disparities in incidence for CSTS of the head and neck.

National Comprehensive Care Network (NCCN) guidelines recommend proper biopsy technique, followed by surgical resection with adequate margins and complete histological analysis for the treatment of CSTS of the head and neck [1]. Disparate access to cancer treatment in general [12,13] and surgical oncology in particular have been previously reported in the literature [14,15]. Patients with CSTS of the head and neck typically present with a skin mass and may present to a physician without an oncologic training. As a consequence, these patients may undergo an unplanned biopsy/surgical excision without attention to oncologic principles [1,2]. Considering surgical resection is the cornerstone for cancer therapy for CSTS of the head and neck, our goal was to explore the disparities in overall survival. We have utilized the National Cancer Institute’s (NCI), population-based Surveillance, Epidemiology and End Result (SEER) database for our analysis.

## 2. Methods

The cohort of cases for the current study was isolated using the NCI’s SEER program [11]. Presently, SEER collects the data from 22 registries covering approximately 48% of the US population [11]. We utilized the ICD-O-3 codes for malignant histologic behavior and primary location to isolate a total of 4253 cases. Histologic recode broad groups included ‘soft tissue sarcoma 8800–8809, fibromatous sarcoma 8810–8839, myxomatous sarcoma 8840–8849, lipomatous sarcoma 8850–8889, myomatous neoplasms 8890–8929, complex mixed and stromal sarcoma 8930–8999 and blood vessel sarcoma 9120–9169’. The number of patients belonging to each broad group are summarized in the Appendix A. In addition, frequency table detailing the number of patients corresponding to individual ICD-O-3 code are also presented in the Appendix A. Given the frequency distribution and for the statistical analysis, we combined the patients with myxomatous, lipomatous and complex mixed and stromal sarcoma categories in a single category. Patients with diagnosis of Kaposi sarcoma 9140.3 were excluded from this analysis, as mainstay of treatment for Kaposi sarcoma is medical therapy and radiation. Primary location included sites ‘C44.0 through C44.4 corresponding to skin of lip, NOS, eye lid NOS, external ear, skin other/unspecified parts of face and skin of scalp and neck, respectively’. The information was extracted from the SEER dataset (18 registries from 2000–2018). Information regarding patient demographics, grade, stage, size, year of diagnosis, surgical and radiation treatment, and overall survival time until death or loss to follow-up was identified. Information regarding socioeconomic status (SES) and insurance status was extracted using the custom SEER census tract level and rurality database from 2000 to 2016 [16]. Patients with no insurance were grouped together with patients on Medicaid. This was done as patients presenting with no insurance to a healthcare facility are enrolled in Medicaid [17]. Small-area SES was analyzed as a composite index calculated by SEER using the method described by Yost et al. [18] Census tract-level SES indicator variables of median household income, median house value, median rent, percentage of the population below 150% of the poverty line, an education index, percentage of the population with working class occupations and percentage of population older than 16 years in the workforce without a job were utilized [18]. The data are presented as quintiles, group 1 representing the lowest SES and group 5 representing the highest SES. Patients with missing data were excluded from each respective univariable and multivariable analysis. Census urban-area-based categorization was used to stratify the cohort in ‘Urban’ and ‘Rural’ groups. ‘All rural’ and ‘mostly rural’ were grouped together under ‘Rural’. Similarly, ‘all urban’ and ‘mostly urban’ were grouped together under ‘Urban’. 

Patient age was converted to a categorical variable (0–14, 15–39, 40–64, ≥65). We chose this stratification to align with adolescent and young adult population demographics being defined at 15–39 [19,20]. Staging categories of local, regional and distant disease were used according to SEER staging system [21]. Tumor size was also converted into a categorical variable (≤5 cm, >5 cm). Size cut off of 5 cm was used as per AJCC 8th ed recommendation for axial soft tissue sarcoma [22]. Surgical procedures were categorized into ‘skin biopsy including Mohs surgery’ and ‘wide excision including amputation’. The primary outcome in the current investigation is ‘overall survival’.

SEER* Stat software (version 8.3.8, NCI, Bethesda, MD, USA) was used to analyze incidence rates which were age adjusted and normalized using the 2000 US Standard population using the dataset ‘18 registries 2000–2018’. Statistical analysis was performed using SPSS Statistical package version 27.0 (SPSS Inc., Chicago, IL, USA). Log-rank test was utilized for categorical values to evaluate the effects of demographic, clinical, pathological, treatment and socioeconomic variables. A multivariable analysis was performed to determine independent predictors of outcome using the Cox proportional hazards model.

## 3. Results

### 3.1. Demographics and Clinical Characteristics

A total of 4253 patients were extracted from the SEER database from 2000–2018. The demographics for the cohort are shown in Table 1. Almost half the patients were diagnosed from 2000–2010 (49.2%). Most of the patients were 65 years of age or older (74.1%). A majority of patients were ‘Male’ (79.9%). The most common racial/ethnic group was non-Hispanic White people (NHW) (89.2%), followed by Hispanic people (5.1%). Most of the tumors were less than 5 cm in size (80.2%) and presented with a ‘localized’ stage (78.6%). The highest number of patients had a ‘undifferentiated’ grade (37.5%). The five most common ICD-O-3 codes include: 8803.3 malignant fibrous histiocytoma (2026, 47.6%), 8832.3 dermatofibrosarcoma (672, 15.8%), 8802.3 giant cell sarcoma (461, 10.8%), 9130.3 hemangiosarcoma (504, 11.9%) and 8890.3 leiomyosarcoma (276, 6.5%). ‘Skin biopsy including Mohs’ (78.1%) was the most common surgical procedure followed by ‘wide excision including amputation’ (21.9%). Only 24.5% of the patients received radiotherapy. An even lower proportion of patients received chemotherapy (4.5%). The majority of the cohort was insured (95.2%) and the highest number of patients were in the fifth quintile of SES (29.8%). We stratified the histological subtypes into broad groups: soft tissue sarcoma (15.8%), fibromatous sarcoma (64.4%), myxomatous, lipomatous, complex mixed and stromal sarcoma (0.7%), myomatous sarcoma (7%), and vascular sarcoma (12.1%). The frequency distribution of individual ICD-O-3 codes is detailed in the Appendix A.

### 3.2. Incidence: Sex and Racial/Ethnic Disparities

The incidence of CSTS of the head and neck was 0.27 per 100,000 persons in 2018 and has increased significantly since 2000 (Figure 1a). The annual percentage change was 1.94 with *p* < 0.05 (Figure 1a). The incidence of CSTS of the head and neck for male patients was 0.51 per 100,000 persons (Figure 1b) as compared to 0.092 for female patients, in 2018 (Figure 1b). Over the study period, males were 5.37 times more likely as compared to females to be diagnosed with CSTS of the head and neck. Non-Hispanic White people had an incidence of 0.325 per 100,000 persons in 2018 (Figure 1c), highest among all racial/ethnic groups. As compared to NHB (incidence 0.078 per 100,000 persons Figure 1c), NHW were 4.62 times more likely to develop CSTS of the head and neck.

### 3.3. Survival and Univariable Analysis

The five- and ten-year overall survival rate for the entire cohort were 0.44 and 0.27, respectively (Table 2). Univariable and multivariable survival analyses are shown in Table 2 and Table 3. On univariable analysis ‘younger’ age (*p* < 0.001), ‘female’ sex (*p* < 0.001), ‘NHB’ racial/ethnic group (*p* < 0.001), ‘well differentiated’ grade (*p* < 0.001), ‘localized’ stage (*p* < 0.001), size of primary tumor less than 5 cm (*p* = 0.04), histologic broad group of myxosarcoma, liposarcoma and complex mixed and stromal sarcoma (*p* < 0.001), surgical excision (*p* < 0.001), surgical procedure ‘wide excision including amputation surgery’ (*p* = 0.025), lack of radiotherapy and chemotherapy (*p* < 0.001) and urban origin of patients (*p* = 0.038) were significantly associated with improved survival. 

Chi-square test of independence was performed to examine the relation between ‘SES’ and ‘surgery’/‘surgical procedure’. In the current analysis, the relation was not statistically significant (data not shown).

Due to the high degree of collinearity between ‘surgical excision’ and ‘surgical procedure’, we ran multivariable analysis with each of the variables separately. The determination of independent prognostic factors using Cox P-H model yielded similar results with either of the variables. We have presented below the multivariable analysis using ‘surgical procedure’ variable (Table 3). 

### 3.4. Multivariable Analysis: Independent Predictors of Overall Survival

On multivariable analysis (Table 3) age group ‘40–64 years’, ‘localized’ and ‘regional’ stage, and ‘fibromatous sarcoma’ histologic group were independent predictors of improved overall survival. SES groups 1, 3 and 4 were independent predictors of worse overall survival. A representative survival curve is shown in Figure 2.

## 4. Discussion

### 4.1. Previous Literature

To the best of our knowledge, this is the first population-based study for patients with CSTS of the head and neck, delineating the survival, prognostic factors and disparities in incidence and overall survival. Previous reports consist of single-center-based data focusing on soft tissue sarcoma of the head and neck [3,4,5]. Patients with CSTS of the head and neck were included as a part of the cohort, however, a focused analysis of survival or prognostic factors was not reported. This was in part due to rarity of the disease; few patients with the diagnosis of CSTS of the head and neck precluded any meaningful analysis. Single-center data are also susceptible to selection bias. Rouhani et al. have utilized SEER database to report on incidence and survival of CSTS in the U.S. [6]. Their analysis focuses on the most common histopathological subtypes; however, an analysis focused on anatomical regions is lacking. Others have used utilized population-based data to report on ‘sarcomas of the head and neck’ and ‘leiomyosarcoma of the head and neck’ [7,8]. These reports lacked the focus on CSTS. The current report is unique as it focuses on the incidence, survival and associated disparities among patients with CSTS of the head and neck.

### 4.2. Sex and Racial/Ethnic Disparities in Incidence

This is the first report of incidence of CSTS of the head and neck and highlights the sex and racial/ethnic disparities in incidence. Males and NHW are ~five times more likely to present with CSTS of the head and neck (Figure 1). Rouhani et al. reported CSTS incidence rate ratio for male to female of 4.7 for malignant fibrous histiocytoma (MFH), 3.7 for leiomyosarcoma (LS), 2.0 for Angiosarcoma (AS), and 0.9 for dermatofibrosarcoma proturans (DFSP) [6]. Our results indicate a slightly higher preponderance of CSTS of the head and neck among males with an incidence rate ratio of 5.37. White people have been found to be more likely to develop MFH, LS and AS, while Black people are more likely to develop Kaposi sarcoma and DFSP [6]. In our analysis, NHW are 4.62 times more likely to develop CSTS of the head and neck as compared to NHB. Of note, we have excluded the cases with diagnosis of Kaposi sarcoma in the current analysis.

### 4.3. Prognostic Factors

Other than ‘year of diagnosis’ and ‘insurance status’, all other factors achieved statistical significance on univariable analysis (Table 2). However, using the Cox P-H model for multivariable analysis, only age group ’40–64 years’, stage other than ‘distant’ and histologic broad group ‘fibromatous sarcoma’ were independent protective factors for improved overall survival. Since a majority of patients with CSTS of the head and neck were older than 40 years of age (92.3%, Table 1); age group ‘40–64 years’ represents improved overall survival with younger age at diagnosis. Age group ‘0–14 years’ was censored in the multivariable analysis and the AYA age group ‘15–39 years’ had only 10 patients (Table 3). This finding is consistent with improved outcomes for younger age previously reported by Peng et al. for head and neck sarcomas [7]. Stage other than ‘distant’ was an independent protective factor for overall survival, as expected, highlighting the importance of early diagnosis and local control to prevent systemic disease. 

The broad histologic group ‘fibromatous sarcoma’ included: 8810.3 fibrosarcoma NOS (0.3%), 8811.3 fibromyxosarcoma (0.2%), 8815.3 solitary fibrous tumor, malignant (0.01%), 8825.3 myofibroblastoma, malignant (<0.1%), 8830.3 malignant fibrous histiocytoma (47.6%), 8832.3 dermatofibrosarcoma (15.8%), 8833.3 pigmented dermatofibrosarcoma (0.3%), and 8836.3 malignant angiomatoid fibrous sarcoma (<0.1%) (Appendix A). Others have previously reported higher survival for DFSP (dermatofibrosarcoma protuberans) and MFH relative to other CSTS, a finding consistent with the current analysis [6]. Malignant fibrous histiocytoma (MFH) is a histologic diagnosis that has undergone considerable change over time. In 2002, the World Health Organization (WHO) declassified MFH as a diagnostic entity [23]. It is now known as undifferentiated pleomorphic sarcoma (UPS) [23]. However, since the terminology ‘MFH’ has been used in the SEER database, we have reported it as such for the sake of consistency and to avoid any confusion.

### 4.4. SES Disparities in Outcomes

Previous reports have highlighted the importance of surgical resection and local disease control in the treatment of sarcoma [3,4,5,7,8]. These studies, however, did not explore the impact of SES. In the current study, ‘surgical resection’ and surgical procedure of ‘wide resection including amputation’ were statistically significant on univariate analysis (Table 2). However, on multivariable analysis, neither surgical resection, nor surgical procedure, achieved statistical significance (Table 3). On the other hand, our analysis highlighted the prognostic significance of higher SES in patients with CSTS of the head and neck (Table 3). This is a unique finding that has not been previously reported in the literature. This finding highlights a potential issue of access to health care in general and surgical oncology in particular among patients with CSTS of the head and neck. Disparities in access to health care [12,13] and surgery specifically, have been previously implicated for patients with cancer [14,15]. The current analysis, however, did not reveal any correlation between SES and surgery/surgical procedure. Multivariable Cox P-H model reveals loss of significance for treatment modalities (surgery, radiotherapy and chemotherapy) when co-analyzed with SES status, raising the possibility of a correlation between SES and treatment modalities collectively.

Limitations of the current study include lack of any information on specific chemotherapy or any other medical therapy in the SEER database. Similarly, no information regarding any medical history, radiological studies or serological work up is provided in the database limiting our analysis. Epidemiological studies comparing SEER areas to non-SEER areas in the U.S. conclude that their age and sex distributions are comparable except that SEER areas tended to be more affluent and more urban than non-SEER areas. Staging can be a potential pitfall in all studies based on the database as lack of any radiological record makes it impossible to verify the stage at diagnosis.

Despite these limitations, the current study constitutes a significant step towards identification of independent factors associated with improved survival and highlights sex and racial/ethnic disparities in incidence and SES disparities in overall survival. The latter finding further highlights a potential issue with access to health care.

## Figures and Tables

**Figure 1 jcm-11-05475-f001:**
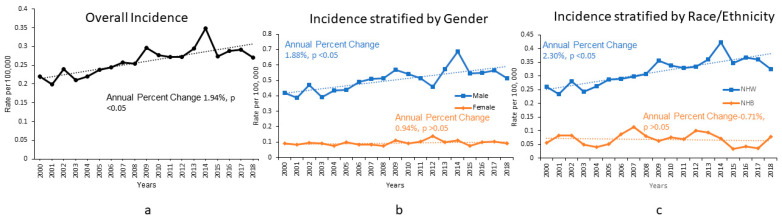
Incidence of Cutaneous Soft Tissue Sarcoma (CSTS) of Head and Neck. (**a**) Overall Incidence. (**b**) Incidence stratified by Gender. (**c**) Incidence stratified by Ethnicity. NHW: Non Hispanic White, NHB: Non Hispanic Black.

**Figure 2 jcm-11-05475-f002:**
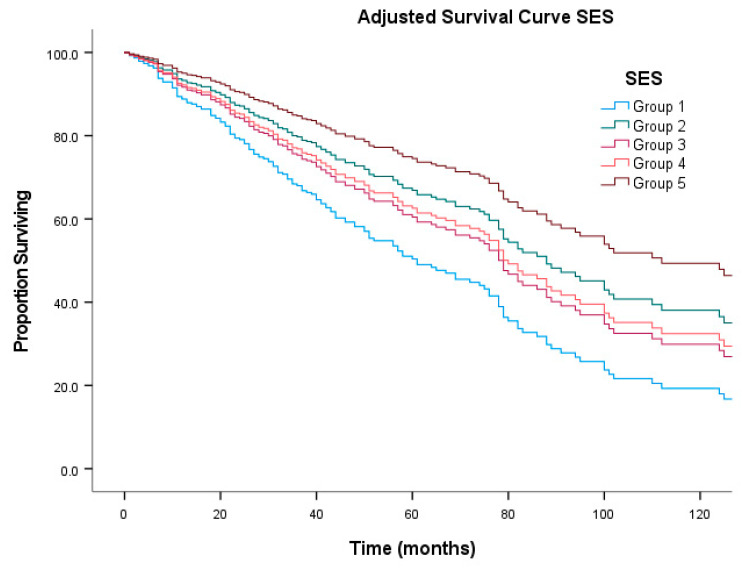
Adjusted survival curve stratified by socioeconomic status (SES).

**Table 1 jcm-11-05475-t001:** Demographics and Clinical characteristics of the entire cohort of Cutaneous Soft Tissue Sarcoma (CSTS) of Head and Neck.

Demographics and Clinical Characteristics of the Entire Cohort		Number of Patients	Valid 100% of Total
Total Patients		4253	100
Age			
	00–14 years	27	0.6
	15–39 years	301	7.1
	40–64 years	763	17.9
	≥65 years	3162	74.3
Sex			
	Male	3398	79.9
	Female	855	20.1
Race/Ethnicity			
	NH White	3643	89.2
	NH Black	119	2.9
	NHAPI	12	0.3
	NHAIAN	98	2.4
	Hispanic	210	5.1
Grade			
	Well Differentiated	80	10.2
	Moderate	182	23.2
	Poorly	229	29.1
	Undifferentiated	295	37.5
Stage			
	Localized	2725	78.6
	Regional	659	19
	Distant	81	2.3
Size			
	<5 cm	1397	80.2
	≥5 cm	344	19.8
Histology			
	Soft Tissue Sarcoma	673	15.8
	Fibromatous Sarcoma	2738	64.4
	Myxo-, Lipo-, Complex Mix Sarcoma	31	0.7
	Myomatous Sarcoma	296	7
	Vascular Sarcoma	515	12.1
Surgery			
	Surgery	3414	81.2
	No Surgery	789	18.8
Surgical Procedure			
	Skin Biopsy Including Mohs’	2666	78.1
	Wide Excision Including Amputation	748	21.9
Radiation Therapy			
	Radiotherapy	1040	24.5
	None	3200	75.5
Chemotherapy			
	Chemotherapy	165	4.5
	None	3509	95.5
Insurance			
	Insurance	2357	95.2
	No Insurance/MedicAid	119	4.8
SES			
	Group 1	351	10.2
	Group 2	581	16.9
	Group 3	659	19.2
	Group 4	822	23.9
	Group 5	1026	29.8
Rurality Index			
	Rural	374	10.8
	Urban	3095	89.2
Year of Diagnosis			
	2000–2010	2093	49.2
	2011–2018	2160	50.8

NH: Non Hispanic, API: Asian Pacific Islander, AIAN: American Indian Alaskan Native.

**Table 2 jcm-11-05475-t002:** Overall survival according to demographic and clinical characteristics (proportion surviving).

Overall Survival According to Demographic and Clinical Characteristics (Proportion Surviving)		5 Years Survival	10 Years Survival	*p*-Value
Overall		0.44	0.27	n/a
Age				
	00–14 years	0.95	0.95	
	15–39 years	0.94	0.91	
	40–64 years	0.78	0.63	
	≥65 years	0.3	0.12	<0.001
Sex				
	Male	0.41	0.23	
	Female	0.56	0.43	<0.001
Race/Ethnicity				
	NH White	0.4	0.23	
	NH Black	0.76	0.64	
	NHAPI	0.54	0.45	
	NHAIAN	0.65	0.65	
	Hispanic	0.67	0.58	<0.001
Grade				
	Well Differentiated	0.52	0.3	
	Moderate	0.43	0.27	
	Poorly	0.26	0.11	
	Undifferentiated	0.31	0.13	<0.001
Stage				
	Localized	0.44	0.25	
	Regional	0.48	0.36	
	Distant	0.16	0.12	<0.001
Size				
	<5 cm	0.43	0.28	
	≥5 cm	0.37	0.24	0.04
Histology				
	Soft Tissue Sarcoma	0.37	0.23	
	Fibromatous Sarcoma	0.51	0.31	
	Myxo-, Lipo-, Complex Mix Sarcoma	0.62	0.51	
	Myomatous Sarcoma	0.36	0.24	
	Vascular Sarcoma	0.2	0.1	<0.001
Surgery				
	Surgery	0.46	0.29	
	No Surgery	0.35	0.21	<0.001
Surgical Procedure				
	Skin Biopsy Including Mohs’	0.45	0.28	
	Wide Excision Including Amputation	0.48	0.3	0.025
Radiation Therapy				
	Radiotherapy	0.4	0.23	
	None	0.45	0.28	<0.001
Chemotherapy				
	Chemotherapy	0.17	0.13	
	None	0.44	0.27	<0.001
Insurance				
	Insurance	0.4	0.33	
	No Insurance/MedicAid	0.51	0.43	0.085
SES				
	Group 1	0.46	0.28	
	Group 2	0.44	0.27	
	Group 3	0.4	0.24	
	Group 4	0.41	0.26	
	Group 5	0.44	0.29	0.01
Rurality Index				
	Rural	0.39	0.25	
	Urban	0.43	0.27	0.038
Year of Diagnosis				
	2000–2010	0.43	0.27	
	2011–2018	0.51	~	0.197

~ Statistic could not be calculated. *p* value shown for Log rank test between variables; Age: *p* < 0.001 only for age ≥65 vs. the rest and for 15–39 vs. 40–64; *p* = 0.923 for 0–14 vs. 15–39; *p* = 0.069 for 0–14 vs. 40–64; Race/Ethnicity: *p* < 0.001 is true only for NHW vs. NHB and NHW vs. Hispanic; *p* = 0.160 for NHW vs. NHAIAN; *p* = 0.051 NHW vs. NHAPI; *p* = 0.740 for NHB vs. NHAIAN; *p* = 0.002 for NHB vs. NHAPI; *p* = 0.283 for NHB vs. Hispanic; *p* = 0.338 for NHAIAN vs. NHAPI; *p* = 0.987 for NHAIAN vs. Hispanic; and *p* = 0.012 for NHAPI vs. Hispanics; Grade: *p* < 0.001 only for Well Differentiated vs. Poorly, Well Differentiated vs. Undifferentiated, Moderate vs. Poorly and Moderate vs. Undifferentiated; *p* = 0.254 for Well Differentiated vs. Moderate; and *p* = 0.259 for Poorly vs. Undifferentiated; Stage: *p* < 0.001 for Distant vs. Localized and Distant vs. Regional only; *p* = 0.109 Localized vs. Regional; Histology: *p* < 0.001 only for Soft tissue sarcoma vs. Fibromatous sarcoma, Fibromatous sarcoma vs. Myomatous sarcoma, and Vascular sarcoma vs. the rest; *p* = 0.003 Soft tissue sarcoma vs. Myxo-, Lipo- and Complex Mix sarcoma; *p* = 0.301 for Soft tissue sarocma vs. Myomatous sarcoma; *p* = 0.123 for fibromatous sarcoma vs. Myxo-, Lipo-, Complex Mix sarcoma; and *p* = 0.012 Myxo-, Lipo-, Complex Mix sarcoma vs. Myomatous sarcoma; SES: *p* = 0.01 only for group 3 vs. group 5; *p* = 0.619 for group 1 vs. group 2; *p* = 0.183 for group 1 vs. group 3; *p* = 0.785 for group 1 vs. group 4; *p* = 0.518 for group 1 vs. group 5; *p* = 0.344 for group 2 vs. group 3; *p* = 0.761 for group 2 vs. group 4; *p* = 0.170 for group 2 vs. group 5; *p* = 0.176 for group 3 vs. group 4; and *p* = 0.217 for group 4 vs. group 5. NH: Non Hispanic, API: Asian Pacific Islander, AIAN: American Indian Alaskan Native.

**Table 3 jcm-11-05475-t003:** Multivariable analysis.

Multivariable Analysis		Number of Patients	Hazard Ratio	95% CI	*p*-Value
Age					
	00–14 years	~	~	~	~
	15–39 years	10	0	0–2.82 × 10^167^	0.946
	40–64 years	50	0.245	0.139–0.433	<0.001
	≥65 years	233		Reference Group	
Sex					
	Male	236	0.706	0.460–1.083	0.111
	Female	57		Reference Group	
Race/Ethnicity					
	NH White	257	1.184	0.571–2.456	0.649
	NH Black	6	0.46	0.139–1.526	0.205
	NHAPI	13	2.057	0.715–5.923	0.619
	NHAIAN	1	1.785	0.182–17.471	0.181
	Hispanic	16		Reference Group	
Grade					
	Well Differentiated	29	0.947	0.522–1.718	0.857
	Moderate	71	0.804	0.515–1.256	0.338
	Poorly	90	0.996	0.679–1.461	0.985
	Undifferentiated	103		Reference Group	
Stage					
	Localized	184	0.256	0.113–0.582	0.001
	Regional	100	0.356	0.159–0.799	0.012
	Distant	9		Reference Group	
Size					
	<5 cm	227	0.804	0.55–1.175	0.259
	≥5 cm	66		Reference Group	
Histology					
	Soft Tissue Sarcoma	73	0.675	0.408–1.116	0.126
	Fibromatous Sarcoma	104	0.587	0.374–0.922	0.021
	Myxo-, Lipo-, Complex Mix Sarcoma	5	0.463	0.131–1.640	0.233
	Myomatous Sarcoma	31	0.639	0.346–1.178	0.151
	Vascular Sarcoma	80		Reference Group	
Surgical Procedure					
	Skin Biopsy Including Mohs’	209	0.882	0.622–1.250	0.481
	Wide Excision Including Amputation	84		Reference Group	
Radiation Therapy					
	Radiotherapy	82	0.861	0.593–1.248	0.429
	None	211		Reference Group	
Chemotherapy					
	Chemotherapy	25	1.599	0.848–3.014	0.147
	None	268		Reference Group	
SES					
	Group 1	34	2.33	1.3–4.178	0.005
	Group 2	50	1.368	0.821–2.279	0.229
	Group 3	57	1.711	1.069–2.738	0.025
	Group 4	76	1.594	1.022–2.486	0.04
	Group 5	76		Reference Group	
Rurality Index					
	Rural	37	0.746	0.438–1.271	0.281
	Urban	256		Reference Group	

NH: Non Hispanic, API: Asian Pacific Islander, AIAN: American Indian Alaskan Native.

## Data Availability

The data used in this investigation are available to public at https://seer.cancer.gov, https://seer.cancer.gov/data-software/ (accessed on 11 October 2021).

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
