# Peer review of "Head and Neck Cutaneous Soft-Tissue Sarcoma Demonstrate Sex and Racial/Ethnic Disparities in Incidence and Socioeconomic Disparities in Survival"

_jcm, 2022, doi:10.3390/jcm11185475_

Round 1
Reviewer 1 Report
Thank you for letting me review this manuscript describing differences in incidence and survival in head and neck CSTS. The results are clearly presented and easy to read. I would appreciate more discussion of possibele reasons for the differences detected.
Author Response
Thank you so much for your time and useful suggestions. We have briefly touched upon the possible reasons behind the differences observed in outcomes, in the discussion section. However, there is limited data available in the SEER database. We are being careful to avoid any over conclusion. Given the currently available data and analysis, our ability to hypothesize a causative association is limited. I hope the reviewer can understand our position.
Reviewer 2 Report
This study utilised the SEER database to describe the incidence and characteristics of patients presenting with cutaneous soft tissue sarcoma of the Head and Neck. The authors describe incidence, demonstrating interesting age, sex and race/ethnicity associations and undertook univariable and multivariable survival analysis, which showed that younger age, locoregional disease and fibromatous sarcoma were associated with better outcomes and lower socioeconomic status with worse outcomes.
Overall I think this is an interesting paper, if niche, and I have the following comments:
1. The authors state that there has been no population based analysis of cutaneous sarcoma of head and neck previously. Whilst there may be no dedicated studies they should reference and discuss studies such as RH Woods et al ANZ J Surg 88 (2018) 901–906 in which a population based study of H&N sarcomas includes clear description of the cutaneous sarcomas and sets this group in context of the whole group of H&N sarcomas. Suggest Checking previous larger studies of H&N sarcoma to see whether any others report on cutaneous tumours as subgroup analysis
2. The authors need to set the context ie in the SEER database cutaneous sarcomas are the commonest soft tissue sarcomas of H&N see Peng 2014 (reference 7). By contrast in the Australian study above, soft tissue was the commonest site .Is there any controversy or misreporting on site?
3. No supplementary material was available to me (referenced in line 81)
4. The commonest ICD codes is a result, not a method (lines 80-84).
5. Why are no NECK anatomical sites included (line 89-90) or is the description not complete in the text? Is this actually an analysis of head and neck cutaneous sarcomas or just head/face?
6. What % of all H&N sarcomas in the SEER database for this time period were cutaneous?
5. Please make it clear in results section that patients < 40 yrs were excluded from multivariable analysis of age as prognostic factor .
Author Response
This study utilised the SEER database to describe the incidence and characteristics of patients presenting with cutaneous soft tissue sarcoma of the Head and Neck. The authors describe incidence, demonstrating interesting age, sex and race/ethnicity associations and undertook univariable and multivariable survival analysis, which showed that younger age, locoregional disease and fibromatous sarcoma were associated with better outcomes and lower socioeconomic status with worse outcomes.
Overall I think this is an interesting paper, if niche, and I have the following comments:
Thank you so much for your time and useful suggestions.
- The authors state that there has been no population based analysis of cutaneous sarcoma of head and neck previously. Whilst there may be no dedicated studies they should reference and discuss studies such as RH Woods et al ANZ J Surg 88 (2018) 901–906 in which a population based study of H&N sarcomas includes clear description of the cutaneous sarcomas and sets this group in context of the whole group of H&N sarcomas. Suggest Checking previous larger studies of H&N sarcoma to see whether any others report on cutaneous tumours as subgroup analysis
Thank you for bringing the publication to our attention. We have included it in the introduction section. All the changes have been tracked. Unfortunately, such a study was lacking for the US population and that is why we have conducted the current study.
- The authors need to set the context ie in the SEER database cutaneous sarcomas are the commonest soft tissue sarcomas of H&N see Peng 2014 (reference 7). By contrast in the Australian study above, soft tissue was the commonest site .Is there any controversy or misreporting on site?
Thank you for bringing this interesting point to our attention. Based on our review of the two publications, Peng et al. appears to have grouped skin with other soft tissue locations in the head and neck. Their primary location ‘skin and soft tissue’ does not represent skin soft tissue primary site. And Australian study has also referred to the soft tissues as the commonest site. It appears that the finding is consistent.
Since, we are focusing on cutaneous soft tissue sarcoma, we feel it is not be relevant to set up the stage regarding soft tissue sarcoma of the head and neck in Australia and the US.
- No supplementary material was available to me (referenced in line 81)
Supplementary material contains a couple of tables with histopathological stratification of the tumors as described in the SEER database. It has been uploaded on the journal website. We are not sure what can we do to make sure that it is available for review.
- The commonest ICD codes is a result, not a method (lines 80-84).
Thank you, we have moved it to the results section. (lines 134-137)
- Why are no NECK anatomical sites included (line 89-90) or is the description not complete in the text? Is this actually an analysis of head and neck cutaneous sarcomas or just head/face?
Thank you for bringing this to our attention. We have updated the description of the ICDO-3 site codes lines 89-90. Skin of neck was included as the primary site and is coded as C44.4 ‘Skin of Scalp, neck’.
- What % of all H&N sarcomas in the SEER database for this time period were cutaneous?
Thank you for an interesting question. However, we did not extract the data for all the H&N sarcomas in the SEER database. We do not have the number available to report the percentage. Since our paper is focusing only on the CSTS of the H&N, we are unable to share this information readily.
- Please make it clear in results section that patients < 40 yrs were excluded from multivariable analysis of age as prognostic factor .
We have not excluded the patient < 40 years from multivariable analysis. Patients in age group 00-14 years were censored in the multivariable analysis. This information is presented clearly in Table 3. Age group 15-39 years was not censored and corresponding statistic has been mentioned.